# Collaboration in health promotion for newly arrived migrants in Sweden

**Sara Svanholm**◎*◎, **Heidi Carlerby**◎, **Eija Viitasara**◎◎

Department of Health Sciences, Mid Sweden University, Sundsvall, Sweden

◎ These authors contributed equally to this work.
* sara.svanholm@miun.se

## Abstract

As a group, newly arrived migrants in Sweden face inequities in health compared to the general population. Successful promotion of population health requires awareness of and focus on health from several sectors of society. In light of this, the aim was to study the views of local authority officials on collaboration in health promotion activities for newly arrived migrants. Data was collected through five focus group interviews with 23 local authority officials working with the integration of newly arrived migrants in the Establishment Program in a municipality or at the Employment Services in northern Sweden. An inductive qualitative latent content analysis was performed, and the analysis showed that the participating officials considered health promotion as desirable in the Establishment Program, but it also raised complex issues within the existing organisations. The officials described unclear roles, but also possible changes to the organisation that would improve the possibility of working to promote health. The present study adds to the relatively limited knowledge of health promotion in integration activities and offers clinical relevance for policymakers through the officials' suggestions for improvements in the Establishment Program. It also raises important questions for further research.

**Data Availability Statement:** All relevant data is included in the article.

**Funding:** The authors received no specific funding for this work.

## Introduction

Health inequities affecting migrants are created in a complex system of policy interaction and actions on social systems, where avoidable differences in economic, social and cultural status cause differences in health outcomes [1]. Therefore, to promote health, actions and considerations of the social determinants of health (SDH) are important [2]. SDH are aspects of society, such as education, economics, working conditions, unemployment, housing, culture and healthcare, that all affect health [3]. The populations' health is thus affected by policies and action in different sectors of society [4, 5] and it is important that those sectors are aware of their effect on health as well as their need to take responsibility to promote health through governance, policy, and action [6].

Health promotion (HP) as defined by the World Health Organization is 'the process of enabling people to increase control over, and to improve, their health' [7], and an intersectoral approach is one way to view how different sectors work together to promote health [6, 8, 9]. It

**Competing interests:** The authors have declared that no competing interests exist.

is a strategy where health is put on the agenda outside of the health sector, such as in the municipalities or in the employment services [10, 11], and it can be used effectively to target inequities in health [8].

## Health and migration

The health of migrants is affected throughout the entire migration process, from pre-departure to post-migration and when settling in the destination country [12, 13]. International research has indicated that migrants often have relatively good health when they arrive in the destination country, and that they do not have higher overall mortality rates than the general population [14]. This has been called the 'healthy migrant effect', a commonly used theory that states that migrants are healthier than the population in the country of origin as well as population in the country of destination [14, 15]. However, this has been questioned in recent research, especially when looking only at refugees [15]. A Danish study showed that five years after arrival, refugees and their family members had a higher disease burden than the Danish population at large [16]. Similarly, a Swedish longitudinal study showed that non-western migrants had an increased disease burden six years after arrival, at the first follow-up [17]. Further, international research has shown that migrants have an increased risk of inequalities in the social determinants of health (SDH), such as lower income, no or worse conditions of employment, lower quality housing, insufficient skills in the host language and risk of discrimination [14, 18]. These are factors that affect migrants' social position, power, material resources and also increase the risk of ill health, especially ill mental health [18]. Also, in Sweden, research has shown that refugees and migrants from low- and middle-income countries have a higher risk of poor mental health because of low socioeconomic status and negative effects on the SDH after migration [19, 20]. Individuals who have migrated to Sweden form one group facing health inequities [21]. Apart from the structural determinants affecting migrants' health, a Swedish survey showed that they have an increased risk of behaviours that effect health negatively, such as a higher rate of smoking and physical inactivity, compared to the native-born population [22]. Early preventive health efforts targeting migrants' health could benefit both migrants and the society in general [16]. An intersectoral approach is both useful and important when tackling the health risks of people who have migrated [23].

## The Swedish context and the establishment program

Sweden received an increased number of asylum seekers in 2014 and 2015. The influx has since slowed, but because of the long administrative processes, they received their residence permits in 2016 and later, causing the numbers of newly arrived migrants to increase even as the influx of asylum seekers slowed [24]. In 2017, approximately 2,700 (11,4%) of the asylum seekers who received residence permits were placed in a municipality in northern Sweden [25]. In Sweden, refugees, persons of subsidiary protection as well as their family members who have arrived through family reunification ("newly arrived migrants") have the right to participate in a two-year-long introduction program to Swedish society as soon as they receive a residence permit. The Establishment Program [EP] is a national civic orientation program focused on employability. The Employment Services has the overall organising responsibility, and the municipalities, among other authorities, are largely responsible [26]. From 2010 to 2017, the program included an individual introduction plan with civic orientation, Swedish language classes and other courses to enhance employability [27].

The specific context is important for collaboration [28], and previous research on integration in Sweden have shown that local policymaking is affected by the structural conditions of the municipalities, like economical, demographic, and political factors [29]. It is therefore

important that research regarding integration reflects the whole country. Carlerby and Persson [30] describe a lack of studies on health promotion and newly arrived migrants outside of metropolitan areas in Sweden. This is especially true regarding the northern parts of Sweden, where several municipalities have common features in that they are sparsely populated areas [31], but have received relatively many newly arrived migrants in relation to their population size [32]; this cause specific challenges in integration process of newly arrived migrants. A region in northern Sweden was therefore selected for the study [30].

Considering the actions on structural SDH, such as work and education, the EP is an interesting arena for promoting migrants' health. Previous results from the project this study stem from indicate that authority officials in their work with clients reflect over other health promoting factors such as social networks, societal participation, family relations, physical activity and economic situations [30]. Actions on both structural SDH as well as individual factors such as lifestyle are accommodated within the officials' regular work in the EP, although they are not always framed as health promoting activities. Carlerby and Persson [30] described the role of authority officials in HP activities for newly arrived migrants as largely unknown. The aim of the present study was therefore to study the views of local authority officials on collaboration in health promotion activities for newly arrived migrants in the Establishment Program.

## Method

The current study was part of a larger project called Establishment with a Health Perspective [EHP], a project by Mid Sweden University and the County Administrative Board [30]. The EHP followed the APO-audit [33], and a mixed method approach to increase the understanding [34] of how local officials define and reflect on HP activities in their everyday work in the EP. In short, the project first collected quantitative data using two different methods: a survey focused on responsibilities for health promoting and then an audit-registration of authority officials' reflections on health promotion in their meeting with clients. This was followed by qualitative data collection using focus group interviews. Last, the results were reported back to the organisations to facilitate continued improvement in the quality of authority officials' health promoting work. The entire project has been previously described in report form [35] as well as in article form [30].

### The participants

All officials from the municipalities and the Public Employment Services who worked in the EP were invited to participate in the EHP. Of those, four municipalities were chosen to be included in the focus group interviews to ensure a completeness in answers, but also to consider the available time and resources. The participants were chosen to ensure representation of the region's three types of municipalities: large coastal town, coastal community and rural municipality. In total, 23 officials were included in the focus group interviews. The participants were working in the EP at a social office in municipalities (six participants) or the Public Employment Services (17 participants). Out of the officials, 14 were women and nine were men [30].

### Data collection

Data used in the current study were data collected by focus group interviews, performed to obtain diverse perspectives on HP activities. The focus group interviews followed an interview guide, which was formed by the head researcher (HC) from results in a registration of reflections on HP activities and with input from a reference group of experienced authority officials. The key questions focused on how the officials reflected upon HP activities as well as how they

viewed HP activities in regards to factors such as collaboration, location differences, and accessible resources. A pilot focus group interview was conducted with two individuals with personal experience working with newly arrived migrants. They were recruited from a local non-governmental organisation working to support migrants. An additional research fellow (EV) also participated. No major changes were made after this, see final version of interview guide in supporting information (S1 File).

Following the audit method, the participants were first presented with the preliminary results from the early analyses of reflections of HP activities and quantitative data [33, 36]. The data presented were those health promoting activities the officials had reported to reflect upon in meetings with their clients [30], so as to illuminate the HP activities in their everyday work. Then the focus group interview took place. The head researcher conducted the interviews, and an official from the County Administrative Board, an administrative partner in the original research project, also attended. There were five focus group interviews with three to seven participants in each group. Participants were grouped together based on the geographical placement of their work. The interviews took 38 to 70 minutes, saturation of data was reached and the interviews were recorded and transcribed verbatim by the head researcher [30].

## Data analysis

A qualitative latent content analysis, as described by Graneheim and Lundman [37], with an inductive approach [38] was used to analyse the data. The transcribed material was read through several times by the first author (SS) to get a better understanding of the text. The material was then divided into meaning units. The parts of the material that did not correspond to the aim of the study were removed at this stage of the analysis. The meaning units were condensed, labelled with a code corresponding to the interpreted meaning of the content, grouped together into sub-categories and grouped together into categories and abstracted to one theme. During the process of coding, the material was compared to the whole text several times to ensure the created categories were in line with the data [37]. The quotations were translated from Swedish to English by the first author.

## Ethical considerations

The Mid Sweden University's ethics committee (Dnr: MIUN 2016/656) reviewed the EHP. The project and this study were designed and conducted according to the Declaration of Helsinki Ethical Principles for good research [39]. All participants were informed about the research project, its aims, and the prospect of the results being published; they were assured that their participation would remain confidential. All participants consented to be part of the research project verbally. Consents were noted by both the head researcher and the official from the County Administrative Board.

## Results

A theme present throughout the material was that officials working within the EP viewed collaboration as something complex, but desirable to promote the health of newly arrived migrants. 'Unclear roles' and 'changes to structures' were the two categories identified, (see Table 1). The officials described how an organisation that offers clearer roles and structures would improve collaboration, which in turn was raised as an important factor when working with HP, mainly so their clients would not lose time during the short EP. As a participants explained: 'Why [collaboration is important]? Because otherwise, our newly arrived migrants lose!' (Focus Group 5).

**Table 1. Results of content analysis.**

| Theme | Collaboration is complex but desirable in health promotion activities for newly arrived migrants | |
|---|---|---|
| **Categories** | Unclear roles | Changes to structures |
| **Sub-categories** | Lacking structures | Gathering everyone |
| | Limited knowledge of other organizations | Create one way in |
| | Others' responsibility | One contact person |
| | The own organization's work | Make information available |
| | Difficulties referring | |

## Unclear roles

The officials described a current complex situation regarding collaboration in HP activities among the involved actors. The numbers of actors in the EP, which had grown during the last few years, was described as having increased the complexity of operation:

> I used to be the only one in the municipality and then the collaboration was with one other person. There weren't so many others involved. It wasn't so much of a problem and we didn't need policies, we only needed for the two of us to click (Focus Group 2).

Unclear roles affect the respective officials' views on missions and responsibilities. The officials described a lack of structures in the collaboration, involving both economic factors and guidelines from policymakers and employers, but especially on a more direct, daily collaboration between officials:

> We in the Employment Services have a responsibility to collaborate and we're supposed to encourage and support other authorities, and that's fine when it comes to collaboration [on higher levels], but when it comes to us officials, we're lacking the right tools to collaborate. (Focus Group 5).

Officials described several different aspects of unclear roles in their collaboration. Limited knowledge of how other authorities worked was one aspect. This was both in regards to structures of the organisations and, their responsibility and missions in HP in the EP. One participant described how a lack of personal experience of collaboration with certain authorities did not allow knowledge of how that authority worked, but got informed during the focus group interview: 'I've no experience, I cannot answer the question. . . I thought: is it like that? That's good!' (Focus Group 2).

Limited knowledge of other organisations was also visible when officials made the wrong assumption about other officials and organisations' work. For instance, Participant A in Focus Group 3 said: 'You can't even book a time at the primary healthcare centre in their reception, you need to call them. . .'. Participant B responded: 'We looked that up, they said you could come and book a time in the reception'.

It was clear in the discussions that the officials wanted to learn more about each other, especially regarding how the other officials worked to promote their clients' health. This was also in regards to organisations not included in the interviews, like the Migration Agency, healthcare services, and non-governmental organisations [NGOs].

The differences in roles and the respective authorities' responsibilities were present in the officials' descriptions. The different authorities' work, roles and responsibilities were discussed as separate from each other. The focus in those instances was on what other organisations

should be doing, rather than on collaboration. This was present especially in the discussion on the healthcare services and NGOs where the officials thought others should take greater responsibility in HP activities. One member of Focus Group 5 said: 'I think that maybe the NGOs need to do more outreach work'. Officials also discussed their part in collaborations with the healthcare sector and said that they need to improve their part of the collaboration between the organisations. Concerning collaboration with NGOs, officials raised the fact that they were not used to collaborating, so there was a risk that the officials forgot them when there were opportunities for collaboration. One participant in Focus Group 1 said: 'There's a tradition of collaboration between authorities that aren't there with NGOs, so then you risk forgetting about them because you don't think about NGOs in the same way'. The officials also described difficulties regarding referring their clients between the involved organisations. They pointed out that professional secrecy was not a problem, but they described situations where newly arrived migrants were referred back and forth between the organizations without receiving help. They described a risk for their clients to develop ill health, as well as losing time in the EP because of this. One participant of Focus Group 3 said: 'We get a lot thrown at us. It's difficult to know what to do with it all. We direct the individuals somewhere, but then we get them back after a while. No one did anything. No one took that ball.'

This was especially the case when the primary care and other healthcare organisations, where officials described wanting to refer their clients, but the healthcare did not do the same evaluation or it was difficult to reach healthcare organisations.

## Changes to structures

The participants described various aspects regarding how changes to structures would have an impact on their collaboration in HP activities. They voiced wishes to gather the different organizations, physically or organizationally, in mutual co-operation. They described gathering the different organisations at the same geographic location could be a possible solution to facilitate communication not only between the involved authorities but also with their clients: 'I would say that the best case scenario would be that we had an establishment centre with all organizations gathered under the same roof within close proximity for the clients'. (Focus Group 1).

The fact that they did not just want to sit together, but also work together was stressed: '. . .not just sit together, but work together from different departments'. (Focus Group 3).

Another aspect they voiced regarding structures was their wish to have a designated contact person at each organisation as a one way into the organisation or have a designated coordinator in one organisation who would handle all communication. This was proposed to solve the difficulties of not knowing who and how to contact other organisations:

Yes, that's really important. I think they're getting the hang of it within the healthcare now, they have a person, a coordinator, whom you can contact. So I think it's getting easier, you know whom to contact. (Focus Group 2).

Further, the participants explained that a joint use of resources would save both resources and simplify all officials' work. This, for example, by having one coordinator working with all organisations in the EP. They also saw it as desirable for this coordinator to have sufficient language skills to communicate with the clients without a translator. This, the officials explained, would improve the collaboration and contact with the clients as well as between authorities:

We could use the resources we have better. If we map them I think we could get far. We could divide the work so that we don't have to do double the job, we could help each other

[across authorities] instead of buying resources or services from someone else. We could put that money into something else. (Focus Group 4).

Accessible information was another aspect raised as important, both within and between the organisations, as were translated documents for their clients. They said that if all the officials had the same information it would decrease the risk of giving misinformation to the clients: 'There should be a communal collection of documents, that we could all use, like translated documents too.' (Focus Group 4).The transfer of information to and from NGOs was of special concern regarding this. The officials declared that they had little to no information about NGOs to give their clients. Their desire was to have access to translated information brochures they could hand out, regarding other authorities and NGOs and other initiatives in the civil society to support their clients in navigation between authorities and organisations in HP activities.

## Discussion

This study was based on the results of the EHP project, where collaboration was shown to be an important factor for authority officials' work in HP activities [30]. The aim was to study the officials views on collaboration in HP activities. The results showed that even though the officials viewed collaboration as complex, they described it as valuable and necessary in their work to promote the health of newly arrived migrants. That collaboration is important in HP is in line with previous research [6] and the aspect of collaboration was raised already in the Ottawa Charter [40]. This study adds important knowledge regarding collaboration in HP within authority sectors working with integration, something that has previously been sparsely researched.

The EP as well as the work and responsibilities of the involved authorities are defined by national regulations [41] however, this did not seem to carry over to the HP activities that the officials performed in their everyday work with clients. The first category, 'unclear roles' contained descriptions of the complexity of these unclear roles for the involved authorities and officials in regard to HP in the EP. Several competencies and authorities are working with newly arrived migrants' integration into Swedish society. Belonging to different authorities was indicated as a barrier to collaboration for the officials in the present study. Corbin, Jones and Barry [10] showed through a literature review that to achieve collaboration, representation from all sectors need; agreed upon or discussed; aim, objectives, roles, and responsibilities. A shared vision is important. In the present study, the officials described shared goals in that they wanted healthy clients who were employed or in an educational setting by the end of the EP. Their views on their roles and responsibilities were not as shared. Although they have a mandate to collaborate, and policies are in place, which is important for collaboration in HP [11, 42], there are complexities and difficulties within the collaboration process. The officials' respective roles and responsibilities regarding HP were not known to the officials, not their own role or others'. Generally, roles and responsibilities in intersectoral collaboration in HP are important for successful health outcomes for the target population [10]. Previous research has shown that the view of being responsible for population health is important for HP [6]. However, research regarding intersectoral collaboration in HP in education, has shown that it can be more important for HP activities to be in line with the organisations' core responsibilities and missions, than that their goals align [43].

Previous studies on the EP have shown that the division of responsibilities between authorities lacks clarity [44], which the present study supports in regard to HP. According to Brännström, Giritli-Nygren, Liden and Nyhlen [45], who interviewed formerly newly arrived

migrants in Sweden about their experience in the EP, migrants have difficulties separating the different authorities and contact with authorities affect migrants' everyday lives as well as their future view on and trust in authorities. This is important since social trust affects both self-rated and objective health [46]. Research show that officials are important for newly arrived migrants' experience in the EP and that officials can contribute to their health [47], and feeling of support [45]. Functioning collaboration is important for both migrants [45, 47] and officials [30]. In the present study, this was visible in the desire for a positive working collaboration. They also repeatedly explained how important that was for their clients.

The second category, 'changes to structures', contained the officials' descriptions of factors that could improve collaboration. Many of the solutions and desire for changing the structure had to do with a wish for the officials to facilitate communication. Corbin, Jones and Barry [10] have shown, through a review of the literature, that communication is important for collaboration in HP. In the present study, there was a link between how the officials described their lack of knowledge of how other authorities functioned, their different roles and the need for structures to facilitate communication. The need to have a clearer structure of whom and how to contact each other could be a symptom of this lack of knowledge. Their desired solution was a more integrated organisation, in contrast to the present separated sectors. This is something that Rantala, Bortz and Armada [8] describe in their review of case studies as an integrated collaboration, with a partial joint organization and resources, which leads to efficient collaboration in HP.

It was clear from the officials' descriptions that increasing numbers of partners within the collaboration increase the complexity. This was especially described as a problem with healthcare services which was also an issue raised in previous research [44]. Research has also shown that it is important for the right sectors to be involved and active in the collaboration [10], and this is especially complex in the healthcare sector. The increased difficulties in communication with the healthcare sector are in large parts explained by the complex system of patients choosing their primary care, causing many actors to be involved in the EP. This can also explain the healthcare sectors' relatively small responsibility in the EP, which is in contrast to how de Leeuw [6] describes that, traditionally, the healthcare sector has a prominent role in, and large responsibility for, intersectoral collaborations to promote health. Furthermore, from results of a review of case studies of intersectoral action for health, Rantala, Bortz and Armada [8] described an increase in complexity when the organisations are from different organisational levels in a society, which is the case in the present study. Employment Services is a national agency with local offices that follows national laws, whereas the municipality acts on the local level only and the healthcare sector operates on a regional level. How this affects health promotion in integration is an important question for further research.

The officials described that improvements in communication should include the perspective of communication and availability for their clients as well. Previous research has shown that language barriers and insufficient skills in the host language are important factors that risk worsening migrants' health [18], so this is an important factor. The officials mentioned especially that translated documents and coordinators able to speak several languages as possible solutions, also having local offices where the officials from all authorities could be easily reached in one location would help. This problem was dependent upon the context and which services were accessible in the area. This differs from area to area and the context of a region in northern Sweden was important. In the rural municipalities, not all authorities had officials on site, but those officials on location more often worked geographically close together. In the larger towns, offices were often spread throughout the town, also creating problems for the clients. A changed organisation with a focus on availability for newly arrived migrants would increase the possibility for them to participate and have a greater control. Social participation

can protect newly arrived migrants from ill health [20], and promotion of empowerment is important for health outcomes [2, 18, 48] and is a core concept in HP [7]. Health awareness within the EP is therefore important, and as found in this study, authority officials have a great chance of working with HP.

With intersectoral collaboration, there is a risk that health could be viewed as a means to an end, and therefore risk missing the importance of the structural determinants [49]. This is important to consider in policymaking. In the present study, when discussing action on a local level and the role of officials, a parallel could be drawn to how health issues were viewed as potential threats to employability. The officials described the importance of health for their clients so they could participate fully in the EP. However, since education and work are such important factors for health [2, 18–20] employability, increased language skills and education remain important for the health of newly arrived migrants. The concept of viewing health as more of a resource rather than a goal of its own is important in HP [40] and in line with research within the health field [50]. Therefore, these actions in the EP could be seen as more than a symbolic aim of HP. Sarker and Joarder [51] found that public health experts sometimes fail to recognise the achievements of other sectors on health. Rod [52] found that intersectoral collaborations in HP can sometimes be such a symbolic aim rather than actually leading to a change. It is therefore important to acknowledge the officials' important HP work in the EP. It is also important to remain focused on the structural determinants of health, such as work, housing and education [49] that are the main focus of the EP but to also strengthen the officials view on their own role and responsibility to promote the health of newly arrived migrants.

## Methodological considerations

The focus in the current article has been to describe the research process, the results and the interpretations well enough for the reader to judge the credibility (the recruitment and the participants were described), transferability (the context of the study was described) dependability (the coding example in supplement material was included) and authenticity (quotations were included in the result). By these aspects, the reader can then judge the trustworthiness of the results [53].

In studies about collaboration, it is important to cover more than one angle of a collaboration [8]. In the current study, two perspectives of the collaboration were covered. The two major authorities with the most frequent meetings with the newly arrived migrants in the EP were chosen to be included. However, other officials and organisations are also included in the EP, such as the healthcare sector, and their contribution could have added valuable knowledge to the results. Further research is needed to cover their perspective. Another perspective that was not taken into account in the present study is that there are several levels, apart from the local, of actors in the society that also affect collaboration and how officials work to promote health. The multi-level governance perspective is common in the integration of migrants [54] but has not been addressed in this study, although further research on this important aspect would be valuable.

The study was a secondary analysis, which present limitations in that the focus group interviews were just partly carried out with the main focus on intersectoral collaboration. However, a secondary analysis offered a way to add valuable knowledge from the already collected data [55]. Since health promotion in integration is a relatively sparsely researched area this study has added important new knowledge and raised questions of importance for further studies.

## Conclusions

The present study illuminated how authority officials view collaboration in health promotion in the Establishment Program for newly arrived migrants as complex, but desirable. It has

shown that although authority officials view health promotion and their clients' health as something important, they do not view it as part of their main mission or responsibility to promote it. Their respective roles in health promotion activities were unclear to themselves and to each other. The present study offers clinical relevance for integrating a health perspective in the Establishment Program, but it also raises important questions for further research in regard to integration and health.

## Supporting information

**S1 File. Interview guide.**
(PDF)

## Author Contributions

**Conceptualization:** Sara Svanholm, Heidi Carlerby, Eija Viitasara.

**Formal analysis:** Sara Svanholm.

**Funding acquisition:** Heidi Carlerby, Eija Viitasara.

**Investigation:** Heidi Carlerby.

**Methodology:** Sara Svanholm, Heidi Carlerby, Eija Viitasara.

**Project administration:** Heidi Carlerby, Eija Viitasara.

**Supervision:** Heidi Carlerby, Eija Viitasara.

**Validation:** Heidi Carlerby, Eija Viitasara.

**Writing – original draft:** Sara Svanholm.

**Writing – review & editing:** Sara Svanholm, Heidi Carlerby, Eija Viitasara.

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
