## [Decision Letter · Decision Letter 0]

26 Feb 2020

PONE-D-19-29112

Collaboration in health promotion for newly arrived migrants in Sweden - a qualitative study

PLOS ONE

Dear MsSvanholm, ,

Thank you for submitting your manuscript to PLOS ONE. After careful consideration, we feel that it has merit but does not fully meet PLOS ONE’s publication criteria as it currently stands. Therefore, we invite you to submit a revised version of the manuscript that addresses the points raised during the review process.

Please pay particular attention to addressing the methodological issues raised by the reviewers.

We would appreciate receiving your revised manuscript by 26 March 2020. To enhance the reproducibility of your results, we recommend that if applicable you deposit your laboratory protocols in protocols.io, where a protocol can be assigned its own identifier (DOI) such that it can be cited independently in the future. For instructions see: http://journals.plos.org/plosone/s/submission-guidelines#loc-laboratory-protocols

We look forward to receiving your revised manuscript.

Kind regards,

Rosemary Frey

Academic Editor

PLOS ONE

Journal Requirements:

2. Please address the following:

- Please include additional information regarding the interview guide used in the study and ensure that you have provided sufficient details that others could replicate the analyses. For instance, if you developed a guide as part of this study and it is not under a copyright more restrictive than CC-BY, please include a copy, in both the original language and English, as Supporting Information. In addition, please provide further details concerning the pilot testing of this guide, for example, how many participants were included and from where were they recruited.

- Please provide additional details regarding participant consent. In the ethics statement in the Methods and online submission information, please ensure that you have specified what type of consent you obtained (for instance, written or verbal, and if verbal, how it was documented and witnessed). If your study included minors, state whether you obtained consent from parents or guardians.

Thank you for your attention to these queries.

Reviewers' comments:

Reviewer's Responses to Questions

**Comments to the Author**

1. Is the manuscript technically sound, and do the data support the conclusions?

Reviewer #1: Partly

Reviewer #2: Yes

2. Has the statistical analysis been performed appropriately and rigorously? 

Reviewer #1: Yes

Reviewer #2: N/A

3. Have the authors made all data underlying the findings in their manuscript fully available?

Reviewer #1: Yes

Reviewer #2: Yes

4. Is the manuscript presented in an intelligible fashion and written in standard English?

Reviewer #1: No

Reviewer #2: Yes

5. Review Comments to the Author

Reviewer #1: This study aims to explore the perspectives of local authorities working on a collaboration with health promotion activities in the Establishment Program (EP) for newly arrived migrants in Northern Sweden. Authors suggest an intersectoral approach to health promotion may reduce health inequalities among migrations in Sweden. To achieve their aim, the authors conduct a secondary inductive qualitative content analysis on focus groups with 23 officials (3-7) participants in each group. Findings suggest that EP officials have an unclear conceptualization of their role in providing health promotion activities, and to implement activities, there would need to be structural changes to encourage collaboration. Of particular note, findings point to challenges in developing a coordinated response among multiple agencies and NGOs to meet the full range of needs of newly arrived immigrants to Northern Sweden.

However, this study has the potential to offer insight into the opportunities and barriers to collaboration and implementing health promotion activities into the EP program; currently, the paper lacks conceptual clarity. Issues in the organization of the lit review and methods make it unclear whether this is an evaluation of an existing health promotion intervention, or the authors are exploring the potential of implementing a health promotion activity into EP. If either is the case, there needs to be a stronger argument about the strengths of the EP program as an appropriate point of intervention for health promotion activities. Methods need to be clarified. After a review of the original study, this secondary analysis does not appear to be a duplication – it would benefit from integrating some of the details from the original study in the methods. Of particular note, transparency is needed how this could have affected the research as a limitation and how findings of the data that was offered to the participants prior to the interview.

Major revisions would be needed to accept the manuscript for publication. Further acceptance would need to be conditional that the authors were able to establish the presentation of quantitative results to study participants did not threaten the integrity of the research.

Introduction:

Currently, the order of the key concepts - migration and the social determinants of health, health promotion, and intersectoral approaches – are mixed up. Making it confusing to the reader.

Social determinants of health described in this definition reflect only the structural level of the social determinants of health. Since the paper is on migration and health inequality – the authors should consider framing migration as a social determinant of health and how migration contributes to health inequality.

Limited explanation about how health promotion in the EP program can address these inequalities and why an intersectoral approach is needed.

Avoid the use of words like “good” – describe the status of population health among the ethnic majority in Sweden.

Be careful about generalizations about social-economic status and mental health. There is more evidence trauma experienced during the migration process, and persistent exposure to discrimination are thought to contribute to inequalities in mental health or poor rated health.

The conceptual understanding of the migration health process in this paper is limited. What about the migration health paradox? Newly arrived immigrants often have better health. How could HP activities in the HP program address trajectories of poor health in the country of destination.

More justification is needed on why the authors selected Northern Sweden as the focus of the study.

Methods:

In the first paragraph, it is unclear whether HP is an activity that local authorities already engaged in.

The preliminary quantitative results from the mixed methods study were presented to the participants. This raised methodological concerns about how participants could have been influenced. To amend this problem, the following explanations are needed:

What were the results that were presented?

Given the results that were presented – it is questionable whether this is purely a qualitative study or a mixed-methods study where only the qualitative analysis revised.

Authors cite Graneheim and Lundman – the article specifies decisions in coding regarding latent or manifest content. This needs to be defined.

A citation is needed for the inductive approach as deductive/inductive content analysis is not specified in the Graneheim and Lundman.

Findings:

Line 167- this quote does not seem to match the findings.

Line 162 – is there evidence guessed other’s role incorrectly?

Line 188 – this is an important quote – it would benefit from additional interpretation.

There needs to be clarification on whether there is a clearly defined role for HP activities among these officials. Is the lack of clarity because they do not have an assigned health promotion role? Then this is not a finding. Or is it that they do not know what other actors in the system do?

Conclusion:

This section seemed to be the strongest in terms of contributions to the literature.

Authors highlight gaps in knowledge of HP implementation in Northern Sweden but do not return to it fully in the discussion.

Line 239 – what did it raise that was in line with the Ottawa Charter?

Who is Corbin, Brännström, Rantala, etc.? A brief summary of the study is needed before referring to how this study contributes to their findings.

Other notes:

Copy editing is needed. There are multiple places where authors split infinitives, and the incorrect tense is used.

Avoid beginning a paragraph with the name of the author. There first needs to be a topic sentence.

Reviewer #2: This is an interesting study on collaboration in health promotion for newly arrived migrants.

Remarks:

1. The topic on health among migrants is a large research field, with a great many publications, also in Sweden. The area is very sparsely described, with only a few references, mostly on mental health. Besides, the authors don’t mention the concept “Healthy Migrant Effect”. Thus, a better brief review of possible health problems among migrants is desirable. There are also reports on the health among migrants in Sweden, one older (in Swedish) but still of interest is from Statens Folkhälsoinstitut, 2002:29 (“Födelselandets betydelse”).

2. The topic of intercultural consultations in primary care in Sweden has also been studied, e.g. by Rothlind E. et al in 2018 (actually published in PLoS One). Thus, problems correlated to these intercultural consultations should at least be mentioned.

3. The authors mention some methodological considerations. The head researcher conducted the focus group interviews; did anyone else in the team participate? Do the participants represent different aspects beside the different types of municipalities? Did the study reach saturation of results, or could some aspects have been missed?

6. PLOS authors have the option to publish the peer review history of their article (what does this mean?). If published, this will include your full peer review and any attached files.

Reviewer #1: No

Reviewer #2: No

---

## [Author Response · Author response to Decision Letter 0]

26 Mar 2020

Dear Editor and Reviewers, 

First off, we would like to thank the Academic Editor and the Reviewers for valuable feedback on the article. We are grateful for all the comments and the thorough review of our article. We have processed them all and incorporated almost every point raised in the now re-written article. See our response to the comments in "response to reviewers" document. 

We can see how the article benefitted from all the comments and we are happy to see the finished result. Please let us know if there is something we have misunderstood or could further change in the text. 

Thank you, looking forward to hear from you again. 

Kind regards, 

Sara Svanholm, Heidi Carlerby & Eija Viitasara.

---

## [Decision Letter · Decision Letter 1]

11 May 2020

Collaboration in health promotion for newly arrived migrants in Sweden

PONE-D-19-29112R1

Dear Dr. Ms Svanholm,

We are pleased to inform you that your manuscript has been judged scientifically suitable for publication and will be formally accepted for publication once it complies with all outstanding technical requirements.

With kind regards,

Rosemary Frey

Academic Editor

PLOS ONE

Additional Editor Comments (optional):

Reviewers' comments:

Reviewer's Responses to Questions

**Comments to the Author**

1. If the authors have adequately addressed your comments raised in a previous round of review and you feel that this manuscript is now acceptable for publication, you may indicate that here to bypass the “Comments to the Author” section, enter your conflict of interest statement in the “Confidential to Editor” section, and submit your "Accept" recommendation.

Reviewer #1: All comments have been addressed

Reviewer #2: All comments have been addressed

2. Is the manuscript technically sound, and do the data support the conclusions?

Reviewer #1: (No Response)

Reviewer #2: Yes

3. Has the statistical analysis been performed appropriately and rigorously? 

Reviewer #1: (No Response)

Reviewer #2: N/A

4. Have the authors made all data underlying the findings in their manuscript fully available?

Reviewer #1: (No Response)

Reviewer #2: Yes

5. Is the manuscript presented in an intelligible fashion and written in standard English?

Reviewer #1: (No Response)

Reviewer #2: Yes

6. Review Comments to the Author

Reviewer #1: (No Response)

Reviewer #2: The authors have revised the manuscript in a satisfactory way. I have no further comments or questions.

7. PLOS authors have the option to publish the peer review history of their article (what does this mean?). If published, this will include your full peer review and any attached files.

Reviewer #1: No

Reviewer #2: Yes: Per Wändell

---

## [Editor Report · Acceptance letter]

15 May 2020

PONE-D-19-29112R1 

Collaboration in health promotion for newly arrived migrants in Sweden 

Dear Dr. Svanholm:

I am pleased to inform you that your manuscript has been deemed suitable for publication in PLOS ONE. Congratulations! Your manuscript is now with our production department. 

With kind regards,

on behalf of

Dr. Rosemary Frey 

Academic Editor

PLOS ONE